# Selective amputation of the pharynx identifies a FoxA-dependent regeneration program in planaria

**Carolyn E Adler[1]\*, Chris W Seidel[1], Sean A McKinney[1], Alejandro Sánchez Alvarado[1,2]\***

[1]Stowers Institute for Medical Research, Kansas City, United States; [2]Howard Hughes Medical Institute, Stowers Institute for Medical Research, Kansas City, United States

**Abstract** Planarian flatworms regenerate every organ after amputation. Adult pluripotent stem cells drive this ability, but how injury activates and directs stem cells into the appropriate lineages is unclear. Here we describe a single-organ regeneration assay in which ejection of the planarian pharynx is selectively induced by brief exposure of animals to sodium azide. To identify genes required for pharynx regeneration, we performed an RNAi screen of 356 genes upregulated after amputation, using successful feeding as a proxy for regeneration. We found that knockdown of 20 genes caused a wide range of regeneration phenotypes and that RNAi of the forkhead transcription factor *FoxA*, which is expressed in a subpopulation of stem cells, specifically inhibited regrowth of the pharynx. Selective amputation of the pharynx therefore permits the identification of genes required for organ-specific regeneration and suggests an ancient function for FoxA-dependent transcriptional programs in driving regeneration.

## Introduction

Many organs in the human body have the potential to repair themselves after injury. For example, the hematopoietic system can replenish the entire blood in an animal from a single stem cell after bone marrow transplants (*Lagasse et al., 2001*; *Shizuru et al., 2005*), and hair follicle and epidermis can regenerate in mammals following injury (*Fuchs and Segre, 2000*; *Seifert et al., 2012*). To initiate regeneration, stem cells must sense an injury, proliferate and differentiate appropriately, and replace the missing organs (*Poss, 2010*). Stem cells (and in particular iPS cells) represent enormous potential for developing therapeutic treatments for disease. However, effective implementation of these technologies will require an improved understanding of stem cell activation and regulation in vivo.

Planarians are a classical system for studying regeneration. After amputation, even small fragments can support regrowth of entire animals (*Morgan, 1898*; *Reddien and Sánchez Alvarado, 2004*), indicating that the resident cells have the capacity to self-renew, and can replace all of the missing tissues comprising the animal (including muscle, nervous system, digestive system, excretory system and epithelial cells). This regenerative capacity depends on a population of stem cells termed neoblasts (*Reddien and Sánchez Alvarado, 2004*). These pluripotent cells are constantly dividing, driving replenishment of all cell types during homeostasis (*Newmark and Sánchez Alvarado, 2000*; *Pellettieri and Sánchez Alvarado, 2007*). Upon amputation, neoblasts are stimulated to divide rapidly (*Baguñà, 1976*; *Reddien et al., 2005a*; *Wenemoser and Reddien, 2010*) and begin to differentiate, but how these stem cells are regulated to produce only the tissues that need to be replaced is unclear.

One hypothesis for how stem cells can produce any tissue in the planarian body on demand is that these cells exhibit heterogeneity across the population, in terms of both gene expression and cell cycle status (*Rink, 2012*; *Reddien, 2013*). Heterogeneity has been observed molecularly by gene expression-profiling of neoblasts purified via fluorescence-activated cell sorting (FACS) (*Hayashi et al., 2010*; *Shibata et al., 2012*). Functional evidence of such heterogeneity is supported by single-cell

*For correspondence: asa@stowers.org (ASA); cae@stowers.org (CEA)

**eLife digest** Some animals can regrow whole limbs or organs after amputation. Flatworms called planaria, for example, can regenerate their whole body from small pieces. This remarkable ability depends on neoblasts—a type of stem cell found in planaria that can detect damaged or lost organs, migrate to the site of damage, produce the required cells, and integrate into the remaining tissues. Researchers hope that studying these animals will reveal ways to use stem cells to regenerate injured limbs or organs in humans.

Planaria have been used in many studies of regeneration. However, manually amputating organs from the flatworms is time-consuming and the resulting wounds vary, which makes it hard to compare regeneration between animals treated in different ways.

Now, Adler et al. have developed a new technique for studying regeneration in planaria. Placing the flatworms briefly into a solution of sodium azide causes the pharynx—an organ that is used for both eating and excretion–to drop off. Using chemicals in this way means the loss of the pharynx leaves a uniform wound, with no damage to the adjacent digestive system, and that large numbers of planaria with identical wounds can be produced rapidly. To ensure that treatment with sodium azide did not alter normal regeneration processes in planaria, Adler et al. carried out manual amputation of tissue in sodium azide-treated flatworms; regeneration in these flatworms was identical to that seen in untreated planaria.

Planaria in which the pharynx had been removed by sodium azide exposure showed rapid recruitment of neoblasts to the wound site, where they formed epidermal, muscle and nerve cells, and organized into a functioning pharynx within a few days. Adler et al. then identified 20 genes that were required for various stages of regeneration. These experiments revealed that a transcription factor (a protein that controls gene expression) called FoxA was specifically required for the regrowth of the pharynx. This is a previously unknown function for FoxA.

transplantation experiments in which some, but not all, stem cells can repopulate and rescue animals lacking stem cells (*Wagner et al., 2011*). However, it is unclear what percentage of the stem cell population is in fact pluripotent, or if these cells produce lineage-restricted stem cells. Recent studies have also demonstrated that discrete subpopulations of neoblasts express markers of differentiated tissues (*Scimone et al., 2011*; *Lapan and Reddien, 2012*; *Cowles et al., 2013*). Therefore, cell fate decisions can be established within neoblasts, but how this happens is unknown.

Normally contained within an internal cavity referred to as the pharyngeal pouch, the pharynx protrudes through a ventral opening upon sensing food or prey and ingests food by contractile peristalsis (*Wulzen, 1917*). The planarian pharynx serves as both the entrance and exit to the digestive system and is a complex organ consisting of multiple tissues including neurons, muscle, epithelial cells and secretory cells (*Hyman, 1951*; *Ishii, 1962*; *MacRae, 1963*; *Kido, 1964*). The pharynx is a large cylindrical structure that clearly lacks dividing stem cells (*Hay and Coward, 1975*; *Newmark and Sánchez Alvarado, 2000*; *Orii et al., 2005*). Previous experiments describing de novo pharynx regeneration in head and tail pieces have shown that mesenchymal cells adopt pharyngeal fate prior to accumulation in the nascent pharynx (*Asai, 1991*; *Bueno et al., 1997*; *Cebrià et al., 1999*; *Kobayashi et al., 1999*). These observations suggest that neoblasts respond to the absence of the pharynx and produce cells of the pharyngeal lineage shortly after amputation. Pharynx regeneration is therefore an excellent model for understanding organ regeneration in general, beginning with pluripotent stem cells that differentiate into distinct cell types, which then integrate with pre-existing tissues to form a functional organ.

Here we describe a novel strategy for amputation of a single organ, the pharynx, and for studying its regeneration. Using feeding behavior as a quantitative assay for regeneration, we screened a library of transcripts upregulated during pharynx regeneration. We show that RNAi of these genes causes a wide range of regeneration phenotypes, and that the pioneer transcription factor FoxA, which functions in many organisms to specify endodermal organogenesis, is required for regeneration of the planarian pharynx. Taken together, this novel amputation strategy offers a defined context in which to measure and understand discrete changes in a stem cell population during regeneration.

# Results

## Chemical amputation selectively removes the pharynx

In order to dissect the specific response to organ amputation, we sought to develop a method to selectively remove a single organ. We found that soaking animals in sodium azide for a brief period of time caused the pharynx to be extruded, and then dislodged completely from the rest of the animal following gentle agitation (*Figure 1A,B*). Because this amputation does not require surgery, the wound produced after chemical treatment is indistinguishable between animals. Within several days the pharynx regenerated, as visualized either in live animals or by hybridization with a pharynx-specific riboprobe (*Figure 1C*; *Cebrià et al., 2007*). Other organs, including the gastrointestinal system, were unaffected at a gross morphological level (*Figure 1D*), indicating that sodium azide treatment causes selective amputation of the pharynx without noticeably perturbing other organs. We term this treatment 'chemical amputation'.

The pharynx is ensheathed by a ciliated epithelium that covers layers of muscle, an extensive neural network, and secretory gland cells (*Cebrià et al., 1997*, *1999*; *Okamoto et al., 2005*). Histological

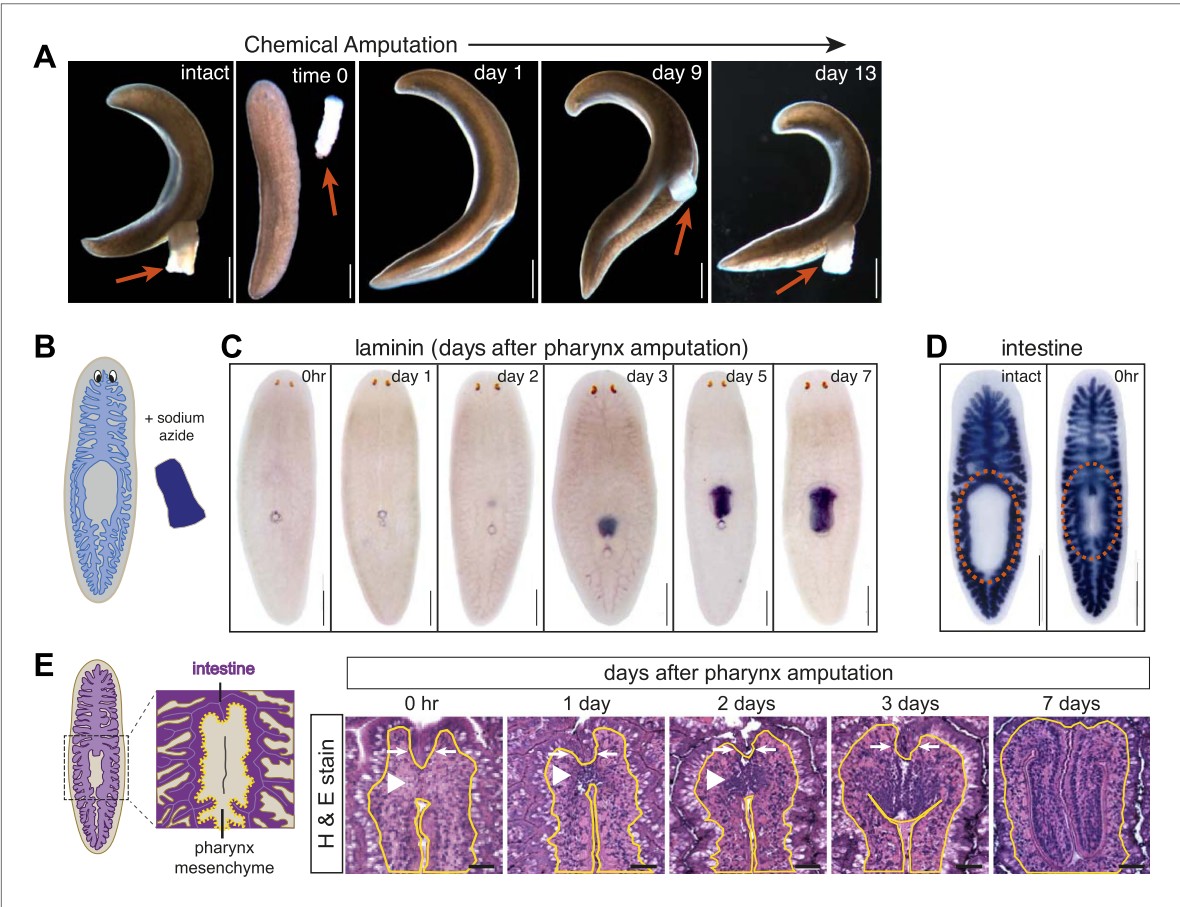

**Figure 1**. Sodium azide selectively removes the pharynx. (**A**) Live animals before and after sodium azide treatment, showing pharynges (arrows). (**B**) Schematic of chemical amputation. (**C**) Pharynx (labeled with *Smed-laminin*) reappears 2–3 days after pharynx removal. (**D**) Intestine (labeled with *Smed-porcupine*) before and immediately after chemical amputation. (**E**) Representative hematoxylin/eosin sections of the regenerating pharynx (white arrowheads). Yellow lines outline mesenchyme and white arrows highlight intestine. Scale bars, **A**–**D**: 500 µm, **E**: 50 µm.

The following figure supplements are available for figure 1:

**Figure supplement 1**. Histological analysis of regenerating pharynx.

**Figure supplement 2**. Effects of sodium azide exposure.

analysis of the anatomical changes that occur during pharynx regeneration confirmed that chemical amputation causes detachment of the pharynx without affecting the intestine (*Figure 1E*, white arrows). Within 24 hr of amputation, small, undifferentiated cells accumulate at the entry to the intestine, within the mesenchyme (*Figure 1E*, white arrowheads). Two days after amputation, these cells begin to organize, characterized by a layer of ciliated epithelial cells, a concentration of neurons, and a subepidermal muscle layer (*Figure 1—figure supplement 1A*). Three days after amputation the lumenal connection to the intestine has been restored, radial symmetry has been re-established, and most genes expressed in the mature pharynx are present, including the secreted Frizzled-related protein *sFRP-1*, ciliary dynein heavy chain *DNAH-β3*, and two members of the Nou Darake family of FGF-receptor-like proteins *ndk* and *ndl-3* (*Rink et al., 2011*; *2009*; *Figure 1—figure supplement 1B*). Therefore, following selective removal of the pharynx, animals regenerate all of the component tissues, in a similar sequence to what has been observed previously for de novo pharynx regeneration (*Bueno et al., 1997*; *Cebrià et al., 1999*; *Kobayashi et al., 1999*).

Even though the regeneration of chemically amputated pharynges appears to proceed normally by all histological and molecular measures, and 100% of animals regenerated pharynges after amputation (n > 1000), we wished to further test whether the brief exposure to sodium azide during chemical amputation might cause secondary effects in regeneration, particularly soon after the treatment. To test the possibility that sodium azide broadly compromised regenerative potential, we performed transverse amputations in sodium azide. After washout, wound healing occurred normally, indicating that animals recovered rapidly from sodium azide treatment. Furthermore, in these regenerating fragments, the mitotic profile triggered by amputation during the early phase of regeneration was indistinguishable from controls (*Figure 1—figure supplement 2*; *Wenemoser and Reddien, 2010*). Altogether, these data demonstrate that sodium azide exposure does not significantly perturb the kinetics of regeneration in general and likely has minimal effects on pharynx regeneration in particular.

## Neoblasts are essential for pharynx regeneration

Exposure of animals to lethal doses of gamma-irradiation completely prevents stem cell division and regeneration (*Bardeen and Baetjer, 1904*). To confirm that pharynx regeneration also requires neoblasts, animals were lethally irradiated (10,000 rads γ-irradiation) prior to pharynx amputation. Radiation completely prevented pharynx regeneration (*Figure 2A*) in 100% of animals (n = 100 animals) indicating that, as expected, stem cells are required for regeneration. Furthermore, lethal irradiation inhibited the accumulation of cells at the wound site 24 hr after amputation (*Figure 2—figure supplement 1*), indicating that the first cells to arrive at the wound site are either neoblasts or their descendants. Similarly, RNAi knockdown of the planarian piwi/Argonaute protein *Smedwi-2* phenocopies radiation by inhibiting stem cell function (*Reddien et al., 2005b*). Indeed, *Smedwi-2(RNAi)* animals failed to regenerate the pharynx (0/33 animals, compared to 24/24 control animals) (*Figure 2B*). These results indicate that pharynx regeneration, like all other regeneration in planaria, depends on functional neoblasts and that large reserves of post-mitotic cells competent to become pharyngeal tissues are unlikely to exist.

In planaria, amputation stimulates two characteristic waves of proliferation: within hours of any wound, mitotic events increase throughout the body, and 2 days later proliferation is localized to the wound (*Baguñà, 1976*; *Wenemoser and Reddien, 2010*). Because chemical amputation produces an internal wound but leaves the epithelium intact, we wondered whether it would elicit similar proliferation kinetics to other types of surgically-induced wounds. We quantified the number of mitoses in the animal during pharynx regeneration by staining planarians with an antibody recognizing phosphorylated histone H3 at serine 10 (*Hendzel et al., 1997*; *Newmark and Sánchez Alvarado, 2000*). In the first 24 hr after amputation, we observed a sharp but transient decrease in overall mitotic activity (*Figure 2—figure supplement 2*), presumably due to the metabolic suppression effects of sodium azide. Overall, body-wide mitotic activity did not significantly change as regeneration progressed (*Figure 2C,D*). However, mitotic activity in the vicinity of the wound site appeared to increase 24 hr after amputation, although this effect diminished as regeneration proceeded. To confirm this observation, we quantified mitoses in each of two defined regions: one around the wound site and one posterior to the wound site (*Figure 2E*, example in *Figure 2C*). Indeed, we found a significant enrichment of mitotic nuclei around the pharynx wound site, indicating that the proliferative response induced by pharynx removal generates a sufficiently powerful signal to induce and maintain proliferation where regeneration is necessary.

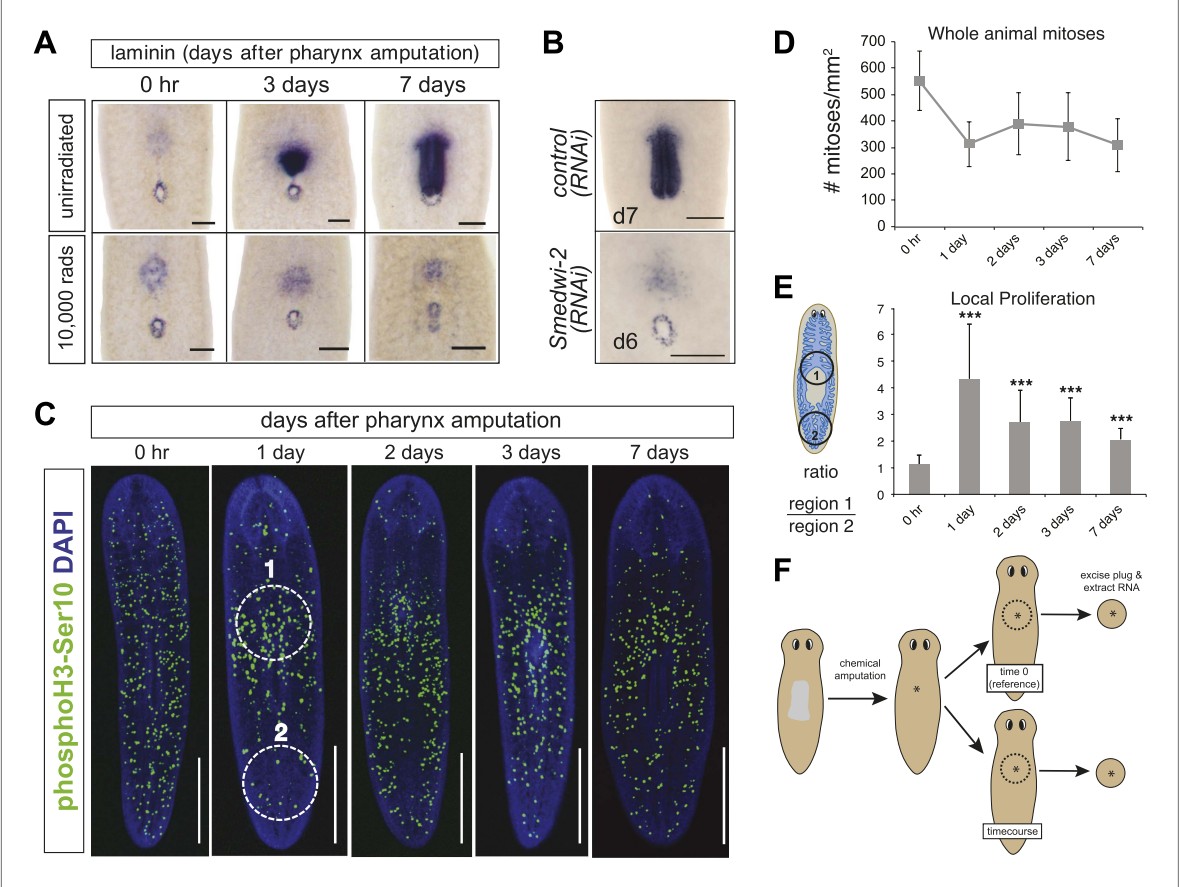

**Figure 2**. Local proliferation of stem cells drives regeneration. (**A**) Irradiated animals fail to regenerate the pharynx (100%; n >50), as indicated by *Smed-laminin* ISH. (**B**) *Smedwi-2(RNAi)* inhibits pharynx regeneration (100%, n >30). (**C**) Representative confocal images of animals during pharynx regeneration, stained with anti-phosphoH3-Ser10. Circles are representative of those used for quantification in (**E**). (**D**) Quantification of phosphoH3-Ser10 staining in whole animals. Error bars = SD. (**E**) Local proliferation measured in two equal-sized circles, (1) centered over the pharynx and (2) centered in the tail as marked in (**C**). Error bars = SD; *** equals p<.0001; significance determined with Student's *t* test. (**F**) Schematic of strategy for expression profiling. Scale bars, **A** and **B**: 200 μm, **C**: 500 μm.

The following figure supplements are available for figure 2:

**Figure supplement 1**. Irradiation prevents accumulation of cells at the blastema.

**Figure supplement 2**. Body-wide mitotic activity after chemical amputation.

**Figure supplement 3**. Validation of microarray by in situ timecourses.

## Gene expression profiling of pharynx regeneration

We sought to define the molecular mechanisms driving pharynx regeneration by expression-profiling experiments. We designed custom oligonucleotide microarrays representing 43,806 predicted *S. mediterranea* transcripts and isoforms from various sources (*Robb et al., 2008*; *Blythe et al., 2010*; *Adamidi et al., 2011*). Based on our observations that pharynx regeneration triggered a localized stem cell proliferative response, we isolated a plug of tissue surrounding the pharynx wound site in order to enrich for those transcripts most directly relevant to this process (*Figure 2F*). We extracted RNA at specific times during a window of time up to 72 hr post-amputation, and compared these samples to plugs isolated immediately after amputation (time 0). Because we aimed to identify transcripts that were essential for the initiation of pharynx regeneration, we first focused on genes upregulated during the first 24 hr after amputation. During this window, 718 genes were enriched upon

pharynx removal (log$_2$ fold change>0.4, adjusted p<0.05) and we cloned 274 of these genes. We also cloned a second group of genes that were significantly upregulated at 48 and 72 hr after amputation, but not prior. An additional set of 82 genes were included based on consistent upregulation at both 48 and 72 hr after amputation, for a total of 356 genes (*Supplementary file 1A*).

To validate our microarray data, we examined the expression patterns of several upregulated genes during pharynx regeneration with whole-mount in situ hybridization. We found that 21/42 of these genes showed distinct upregulation in the area around the regenerating pharynx, as expected (*Supplementary file 1B*; *Figure 2—figure supplement 3*). Interestingly, expression of several of these genes was undetectable prior to amputation, but increased significantly afterward, mimicking the marked upregulation in transcription observed for wound-response genes (*Wenemoser et al., 2012*) and suggesting that chemical amputation does in fact stimulate a wound response. Two genes in this category (PDZ ring finger protein 4 and rhomboid) showed dramatic increases in the region surrounding the pharynx after amputation (*Figure 2—figure supplement 3*), suggesting that these cells may broadly respond to pharynx removal.

## RNAi screen identifies genes required for pharynx regeneration

Because the pharynx is required for ingestion of food, we developed an assay that measures the recovery of feeding behavior after selective pharynx removal as a rapid and quantitative method to gauge the extent of pharyngeal regeneration (*Ito et al., 2001*). When presented with food, planarians normally chemotax toward it and ingest it (*Figure 3A*). If the pharynx is missing or is incompletely regenerated, animals are unable to eat and will not attempt to swim toward the food, implying that the pharynx has a sensory role in stimulating movement towards food. Animals regained the ability to eat 7 days after amputation (*Figure 3B*), indicating that all of the tissues comprising the pharynx were present, functional, and integrated with the rest of the animal by this point in regeneration.

Using feeding behavior as a proxy for pharynx regeneration, we screened the 356 cloned genes by RNA-interference. We considered a 50% defect in food uptake as our initial threshold and identified 20 genes (5.6% of total) that caused reproducible defects in this assay upon knockdown (*Figure 3C*). Analysis of pharynx length by in situ hybridization with the pharynx marker *Smed-laminin* (*Figure 3D,E*) allowed classification of the RNAi phenotypes into groups (see below) with predicted functions during pharynx regeneration. In addition, we measured both the extent of regeneration after head and tail amputation (*Table 1*) and mitotic activity (*Figure 3F*).

The first category of molecules we expected to uncover was general regulators of stem cell function (*Figure 3C*, Stem Cell Effectors), based on the requirement for neoblasts in regeneration. The RNAi screen identified eight genes that exhibited phenotypes consistent with a general function in stem cells. In addition to causing a strong inhibition of feeding and pharynx regeneration (*Figure 3C,D*), knockdown of these eight genes severely compromised regeneration (*Table 1*) and decreased mitotic activity (*Figure 3F*). Moreover, in situ hybridization for these transcripts demonstrated that they were expressed primarily in stem cells (*Figure 3—figure supplement 1*). These genes include ribonucleotide reductase (*Eisenhoffer et al., 2008*; *Böser et al., 2013*), the chromatin assembly factor Rbbp4 (*Bonuccelli et al., 2010*; *Wagner et al., 2012*; *Zeng et al., 2013*), the G1/S-specific cyclin D1 (*Zhu and Pearson, 2013*), the zinc finger-containing protein *zmym-1* (*Wagner et al., 2012*) and the RuvB DNA helicase (*Labbé et al., 2012*). Three other genes in this group have not been previously implicated in stem cell function in planarians. These genes include the DNA licensing factor MCM7, cleavage and polyadenylation specificity factor 3 (CPSF-3), and the Sumo-activating enzyme subunit 2 (SAE-2). Interestingly, SAE-2 was recently identified as a component of the stem cell proteome (*Böser et al., 2013*). Therefore, our screen captured phenotypes for novel stem cell genes, indicating that this strategy can successfully identify genes acting at discrete and early steps in the process of regeneration.

The next group of genes (*Figure 3C*, Specific Effectors) caused profound defects in feeding, but unlike the previous category, most of these animals produced some pharyngeal tissue (*Figure 3C–E*), indicating that pharynx regeneration was either delayed or stalled. The inability to feed was most pronounced following knockdown of the Forkhead transcription factor *FoxA* and the heterogeneous nuclear protein *hnRNPK*. Other genes that caused strong pharynx phenotypes included the poly-A binding protein PABP-2, a component of the ubiquitin proteasome COP9 signalosome complex (COP9), two WD-repeat containing proteins (WDR3 and WDR36), the ER membrane protein jagunal, and decaprenyl diphosphate synthase subunit 2 (DDSS-2). Overall, mitotic activity of the RNAi knockdown animals in this group was comparable to controls (*Figure 3F*), indicating that these

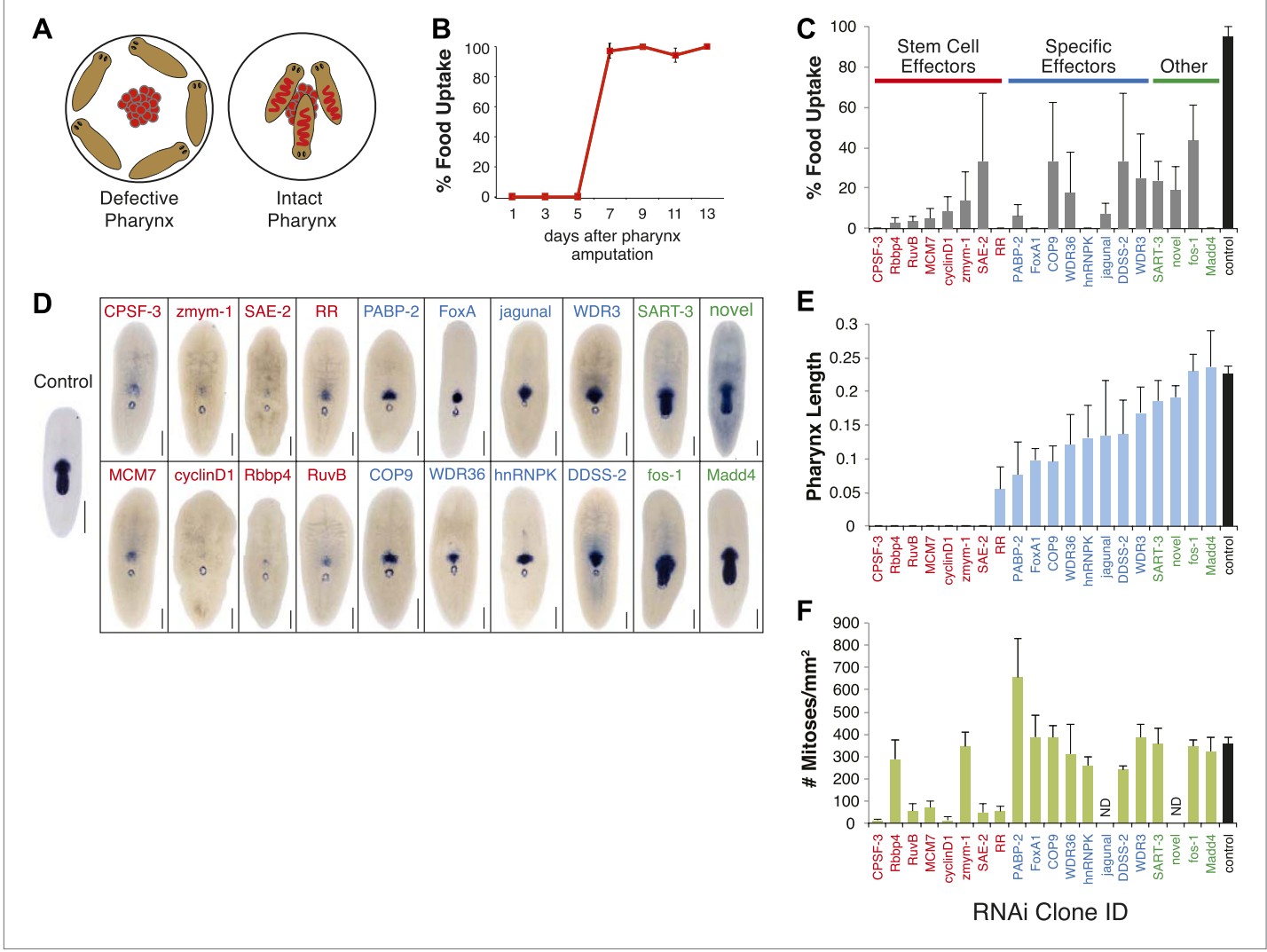

**Figure 3**. RNAi screen for genes affecting pharynx regeneration. (**A**) Schematic of feeding assay. (**B**) Animals recover ability to ingest food 7 days after chemical amputation. For each timepoint, n = 10 animals, repeated in triplicate. Error bars = SD. (**C**) Quantification of feeding behavior of RNAi-treated animals 10 days after amputation. Shown are averages of three independent experiments; error bars = SEM, n ≥30 animals. *Smed-laminin* in situ hybridization shows extent of pharynx regeneration defects in RNAi-treated animals. Scale bars = 250 µm. (**E**) Quantification of pharynx length in RNAi animals 11 days after amputation. For each bar, n = 6–10 animals; error bars = SD. (**F**) Mitotic activity of whole animals 3 days after pharynx amputation measured by phosphoH3-Ser10 staining. Error bars represent SD, and n = 8 animals for each condition.

The following figure supplements are available for figure 3:

**Figure supplement 1**. Candidate gene summary.

genes were unlikely to affect general stem cell function. However, *PABP-2(RNAi)* animals exhibited elevated mitotic activity (*Figure 3F*), a phenotype observed in *EGFR1(RNAi)* and *p53(RNAi)* animals (*Fraguas et al., 2011*; *Pearson and Sánchez Alvarado, 2010*) and reflecting a failure of proliferative control of the stem cell population.

The final category of phenotypes (*Figure 3C*, Other) contains animals that fail to feed but regenerate full-length pharynges (*Figure 3D,E*). To accomplish successful feeding after pharynx amputation, animals need to retain the ability to properly sense food and move towards it. Because our screening strategy relied on feeding behavior, it allowed for the discovery of genes required for motility, chemosensation, or other defects in organ function that do not accompany obvious morphological defects. Indeed, we uncovered several genes in this category, including the immediate early gene

**Table 1.** Summary of RNAi phenotypes

| RNAi ID | Full name | Abbr. | Putative function | Head/Tail Reg | Phx Length | Lysis? | Other phenotypes | *FoxA* expression | Reference |
|---|---|---|---|---|---|---|---|---|---|
| 7D04 | FoxA1 | FoxA1 | transcription factor | phx defective | + | – | dorsal spike | N/A | |
| 5B08 | Mothers against decapentaplegic homolog 4 | Madd4 | Bmp signaling | normal | +++ | – | – | wt | |
| 6F08 | fos/BZIP transcription factor | Fos-1 | immediate early gene | normal | +++ | – | bump over pharynx | wt | Wenemoser et al., *Genes and Development* |
| 5B06 | novel | novel | | normal | +++ | – | – | wt | |
| 1B10 | Heterogeneous nuclear ribonucleoprotein K | hnRNP K | mRNA binding/p53 signaling | small blst | ++ | – | HR | *FoxA* OK; small phx | |
| 5C01 | jagunal | jagunal | ER organization, membrane trafficking | variable | ++ | +/– | | partial | |
| 3C09 | G1/S-specific cyclin-D1 | cyclin D1 | phosphorylates and inhibits Rb | blst (–) | ++ | | | no *FoxA* | Zhu and Pearson, *Development* |
| 2E03 | WD repeat-containing protein 36 | WDR36 | glaucoma disease gene | small blst | ++ | – | | aggregates | |
| 5C12 | Squamous cell carcinoma antigen recognized by T-cells 3 | SART-3 | snRNP complex/interacts with Ago proteins | small blst | ++ | – | | aggregates | |
| 2D06 | WD repeat-containing protein 3 | WDR3 | unclear | small blst | ++ | – | | aggregates | |
| 5G05 | ribonucleoside-diphosphate reductase M2 | RR | deoxyribonucleotide biosynthesis | blst (–) | + | – | | no *FoxA* | Eisenhoffer et al., *Cell Stem Cell* |
| 2A11 | DNA replication licensing factor MCM7 | MCM7 | MCM complex | blst (–) | + | – | | no *FoxA* | |
| 1D01 | Polyadenylate-binding protein 2 | PABP-2 | 3' end processing of mRNA | blst (–) | + | + | curling | reduced FoxA; aggregates | |
| 1C06 | Decaprenyl-diphosphate synthase subunit 2 | PDSS-2 | required for biosynthesis of Coenzyme Q10 | blst (–) | + | – | curling | reduced *FoxA* | |
| 1D07 | COP9 signalosome complex subunit 5 | COP9 | protease subunit of CSN complex | blst (–) | + | + | | *FoxA* OK; small phx | |
| 1B06 | Cleavage and polyadenylation specificity factor 3 | CPSF-3 | 3' end processing of mRNA | blst (–) | – | – | curling | no *FoxA* | |
| 2G03 | Histone-binding protein RBBP4 | RBBP4 | associates with chromatin-regulatory complexes | blst (–) | – | + | | nd | Bonuccelli et al., *J. Cell Sci.*; Wagner et al., *Cell Stem Cell*; Zeng et al., *JCB* |
| 1C07 | RuvB-like 2 | RuvB | DNA helicase involved in Holliday junction formation | blst (–) | – | ++ | | no *FoxA* | Labbe et al., *Stem Cells* |
| 1E02 | Zinc finger MYM-type protein1 | ZMYM1 | cell morphology regulation | blst (–) | – | + | | no *FoxA* | Wagner et al., *Cell Stem Cell* |
| 1E04 | SUMO-activating enzyme subunit 2 | SAE-2 | SUMO ligase | blst (–) | – | + | | no *FoxA* | |

| Phx Length: | normalized to control | Abbreviations |
|---|---|---|
| - | 0–25% | nd = not determined |
| + | 25–50% | blst = blastema |
| ++ | 50–75% | HR = head regression |
| +++ | 75–100% | phx = pharynx |
| | | Reg = regeneration |

fos-1, the SMAD protein Madd4, the RNA-binding protein SART-3, and a novel protein. Following knockdown, these animals regenerated pharynges indistinguishable from controls at the morphological level, and had normal levels of mitotic activity (*Figure 3D–F*). None of these knockdown animals exhibited motility problems, raising the possibility that these genes may be required for sensory function or for proper integration of the newly regenerated pharynx with the rest of the animal.

We then performed in situ hybridizations to determine the distribution of each of these transcripts during pharynx regeneration. Most of these genes demonstrated a striking upregulation in the vicinity of the pharynx as soon as 24 hr after amputation (*Figure 3—figure supplement 1*). Together, these results show that our combined approach of expression profiling followed by RNAi screening successfully identified genes that are functioning at different steps in the regeneration process.

## *Smed-FoxA* is required for regeneration of the pharynx but not other organs

Forkhead transcription factors are critical determinants of foregut development throughout evolution, in both protostomes (*Mango, 2009*) and deuterostomes (*Fritzenwanker et al., 2004*; *Lee et al., 2005*). Expression studies have demonstrated that the planarian homolog of FoxA1 localizes to the nascent pharynx during embryogenesis and regeneration (*Koinuma et al., 2000*; *Martín-Durán and Romero, 2011*), demonstrating that the transcript localizes to developing pharyngeal tissue. In our screen, after three doses of *FoxA(RNAi)*, animals developed a lesion on their dorsal side through which they subsequently ejected their pharynx (*Figure 4A*, 35/50 animals).

To examine in greater detail the anatomical defects caused by *FoxA* knockdown, we analyzed the cellular architecture of the regenerated pharynx after chemical amputation. Three days after amputation, control animals had regenerated the arrayed muscle fibers covered by epithelial cells, along with the ciliated cells that extend to the distal tip of the pharynx (*Figure 4B*). By contrast, *FoxA(RNAi)* pharynges appeared as a disorganized mass of cells, lacking obvious muscle fibers and any epithelial covering (*Figure 4C*), indicating both a failure to produce the cells comprising the pharynx and to pattern them properly. This defect in specification suggests that FoxA is required for coordinating the differentiation of cell types comprising the regenerating pharynx.

Because chemical amputation selectively removes the pharynx without noticeably affecting other organs, we asked whether regeneration of other tissues required FoxA. Tail fragments normally regenerate heads containing a central nervous system, which can be stained with the neuronal markers *Smed-ChAT* and *Smed-PC2*, as well as a new pharynx. *FoxA(RNAi)* tail pieces successfully regenerated nervous system tissue that was indistinguishable from controls, but failed to form a new pharynx (*Figure 4D*). Conversely, head pieces normally regenerate posterior tissue containing two intestinal branches that extend posteriorly around the new pharynx (*Figure 4E*). *FoxA(RNAi)* head fragments had no defect in intestinal growth or patterning (*Figure 4E*), demonstrating that FoxA controls regeneration of the pharynx but not other organs. This is consistent with its role in other organisms as an organ-specific transcription factor.

Transverse amputation of planarians initiates re-patterning of tissues that re-establishes anterior/posterior polarity. Successful regeneration depends on the Wnt pathway and is accompanied by dynamic changes in expression domains of various transcripts (*Gurley et al., 2008*; *2010*; *Petersen and Reddien, 2008*; *2009*). To determine if FoxA is required for establishment of anterior/posterior polarity during regeneration, we examined the patterning of these markers in regenerating fragments. The anterior marker *Sfrp1* and the posterior marker *nou darake-like-3 (ndl-3)* were re-established properly 7 days after amputation in *FoxA* knockdown animals (*Figure 4—figure supplement 1*), indicating that FoxA is not required for specifying expression of these markers.

*Wnt11-5* (also known as *WntP2*) is expressed in a posterior-to-anterior gradient in planarians with its anterior boundary at the pharynx. Initially after removing large regions of the body during amputation, *Wnt11-5* expression is more uniform, but as regeneration progresses the gradient resets to its strong posterior-to-anterior bias (*Petersen and Reddien, 2009*; *Gurley et al., 2010*). This dynamic re-establishment does not depend initially on stem cells, but requires them indirectly for rescaling (*Gurley et al., 2010*). Interestingly, the expression of *Wnt11-5* in *FoxA(RNAi)* tail fragments does not reset the anterior boundary at the pharynx as sharply and reproducibly as controls (*Figure 4F,G*). However, *TOR(RNAi)* animals successfully reset the anterior *Wnt11-5* boundary despite their lack of a pharynx (*Tu et al., 2012*). Therefore, FoxA may participate in specifying the central body region or in regulating stem cells in the vicinity of the pharynx.

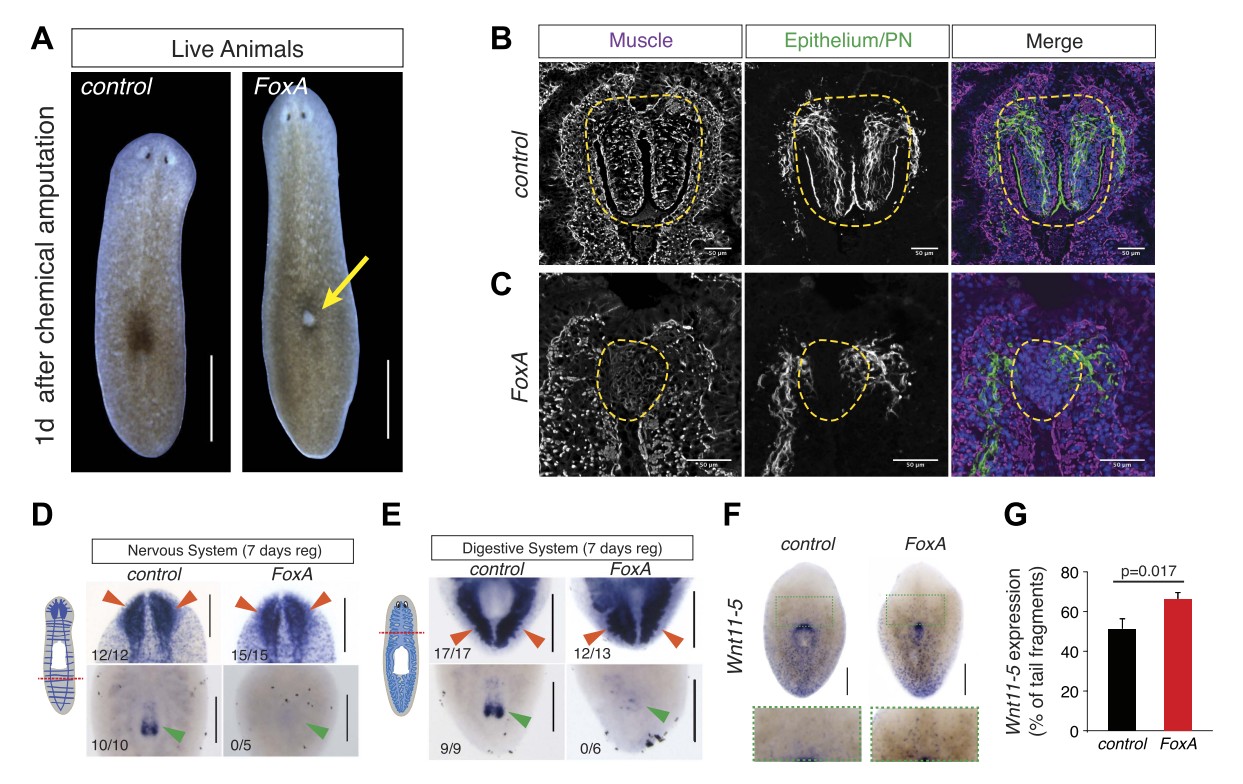

**Figure 4**. *FoxA* is required for pharynx regeneration. (**A**) *FoxA(RNAi)* animals develop dorsal lesions (arrow). (**B** and **C**) Confocal images of cryosections stained with antibodies recognizing muscle (α-Tmus), epithelial cells and protonephridia (α-acetylated tubulin), and nuclei (DAPI). Control (**B**) and *FoxA(RNAi)* animals (**C**) are shown 3 days after pharynx removal. Dashed green lines highlight the regenerating pharynx. (**D**) Tail fragments amputated at dashed red line regenerate brain tissue (*Smed-PC2*, red arrowheads) but not a pharynx (*Smed-PKD2*, green arrowheads). (**E**) Head fragments regenerate posterior intestinal branches (*Smed-porcupine*, red arrowheads) despite the absence of a pharynx (*Smed-PKD2*, green arrowhead). (**F**) Whole-mount ISH for *Wnt11-5* in *control* and *FoxA(RNAi)* tail fragments 7 days after amputation. Green boxes highlight insets shown below. (**G**) Ratio of *Wnt11-5* expression to total length of tail fragment. Significance determined by Student's *t* test. Error bars = SEM. N = 14 fragments. Scale bars, (**A**), 500 μm, (**B** and **C**), 50 μm, (**D**–**F**), 200 μm.

The following figure supplements are available for figure 4:

**Figure supplement 1**. FoxA is not required for anterior/posterior patterning during regeneration.

## FoxA restricts differentiation of stem cells into pharyngeal tissue

In adult planarians, *FoxA* is expressed strongly in the mature pharynx (*Koinuma et al., 2000*; *Umesono et al., 2013*), and in scattered cells in the mesenchyme surrounding the pharynx (*Figure 5A*). These cells were arrayed in a branched pattern reminiscent of neoblasts (*Orii et al., 2005*; *Reddien et al., 2005b*). We observed that after pharynx amputation, *FoxA+* cells accumulated both in the nascent pharynx and tissue surrounding the pharyngeal pouch, adjacent to the wound site (*Figure 5B*, top row). This region contained the densest concentration of *FoxA+* cells 3 days after amputation. Interestingly, this pattern was similar to the expression of 10 other genes from the screen (*Figure 3—figure supplement 1*). These results suggest the possibility that *FoxA+* cells in the mesenchyme may represent pharyngeal progenitors and that our screen uncovered markers for this cell population.

Previous experiments combining partial irradiation of planarians with partial transection of the pharynx have localized pharynx progenitors to the mesenchyme anterior to the pharynx (*Ito et al., 2001*). To determine whether these cells were neoblasts or their descendants that might be incorporated into the regenerating pharynx, we lethally irradiated animals 2 days prior to pharynx amputation. Radiation exposure causes progressive loss of stem cells and their progeny by preventing the production of these cells, making it an effective strategy for identifying lineage relationships in planaria

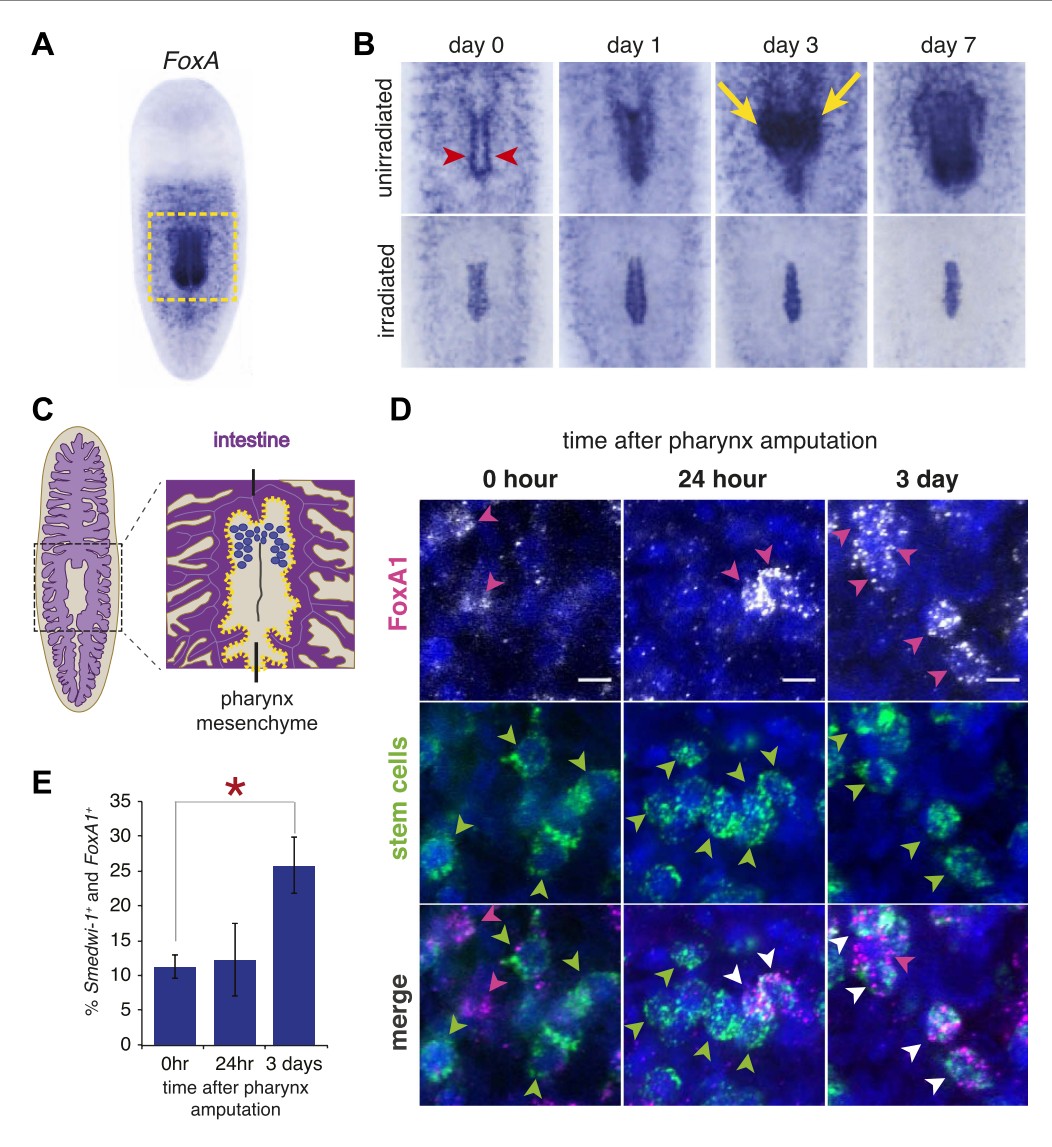

**Figure 5**. FoxA expression in neoblasts increases after amputation. (**A**) Whole-mount ISH for *Smed-FoxA* in intact animals. Boxed region highlights areas shown in (**B**). (**B**) *Smed-FoxA* expression in pharyngeal region during regeneration, in unirradiated animals (top) and lethally irradiated animals (bottom). Yellow arrowheads point to accumulation of *FoxA+* cells in mesenchyme surrounding pharynx and red arrows highlight pharyngeal pouch. (**C**) Schematic of mesenchymal pouch surrounding the pharynx, where *FoxA+* cells concentrate during regeneration. (**D**) Double-FISH with *smedwi-1* and *Smed-FoxA* at different times after pharynx removal. Arrowheads highlight positive cells. Scale bars = 10 μm. (**E**) Quantification of percentage of *smedwi-1+* cells that co-express *FoxA* during regeneration. For each timepoint, n = 100–150 *smedwi-1+* cells.

(*Eisenhoffer et al., 2008*). Radiation completely inhibited accumulation of *FoxA*[+] cells and caused a depletion of the mesenchymal cells surrounding the pharynx (*Figure 5B*, bottom row). Expression of *FoxA* persisted for several days in the epithelial cells lining the pouch and in the central body region. This suggests that pharynx progenitors are neoblast-derived, *FoxA*[+] cells; it also demonstrates that these progenitors localize to the regenerating pharynx rapidly after amputation.

Several transcription factors marking differentiated cell types including photoreceptors, protonephridia and neurons have been shown to initiate their expression in *smedwi-1*[+] stem cells (*Lapan and Reddien, 2011*; *Scimone et al., 2011*; *Cowles et al., 2013*; *Currie and Pearson, 2013*), suggesting that in planaria, lineage decisions can be made within the dividing stem cell population. The expression

of progenitor markers within the pluripotent stem cell population is indicative of heterogeneity of the neoblasts and is thought to contribute to the ability of planarians to regenerate all of their organs equally well (*Reddien, 2013*). Based on our observation that *FoxA* marks a population of irradiation-sensitive pharynx progenitors, we wondered if any or all of them were *smedwi-1⁺* stem cells. We performed double fluorescent in situ hybridization with *FoxA* and *smedwi-1*. Although *FoxA* is expressed relatively weakly compared to *smedwi-1*, in intact animals we found a subset of *smedwi-1* cells that also expressed *FoxA* in the vicinity of the pharynx (*Figure 5D*). This result is consistent with FoxA being present in irradiation-sensitive stem cells.

One of the unanswered questions in planarian biology is how pluripotent stem cells sense the absence of particular organs and how they mount the appropriate regenerative responses to replace those organs. Pairing selective amputation of the pharynx with expression of progenitor markers in the stem cell population allows us to begin addressing this question. We quantified the percentage of *FoxA⁺smedwi-1⁺* cells following pharynx amputation. Indeed, following pharynx removal, the percentage of *FoxA⁺smedwi-1⁺* cells increased significantly, with the peak occurring 3 days after pharynx removal (*Figure 5D,E*). This result demonstrates that a specific, *FoxA⁺* portion of the stem cell population responded to pharynx removal. Interestingly, when small fragments lacking a pharynx initiate regeneration (i.e., from head or tail amputation), they rapidly increase *FoxA* expression in the pharynx rudiment (*Koinuma et al., 2000*). Therefore, mesenchymal cells, even those residing in the anterior or posterior extremes of the animal, can be stimulated to express *FoxA* when regeneration of a pharynx is required.

## FoxA regulates differentiation of neoblasts into pharyngeal tissues

In *Caenorhabditis elegans*, *FoxA* is expressed in all pharyngeal precursors during embryogenesis (*Mango et al., 1994*). By contrast, in adult animals the expression fades, and *FoxA* is preferentially expressed in the intestine (*Panowski et al., 2007*). Embryonic expression studies in flatworms suggest that *FoxA* may not be expressed throughout the definitive organ once development is complete (*Martín-Durán et al., 2010*). To explore whether planarian *FoxA* is expressed throughout the regenerating pharynx, we performed double fluorescent in situ hybridization with markers for known subsets of pharyngeal cell types (muscle, neurons and epithelial cells). 3 days after amputation, when the pharynx has regenerated its cylindrical structure with regularly arrayed muscle fibers and concentrated neurons at its distal tip (*Figure 4B*), *FoxA* was strongly expressed in the epithelial cells and only weakly in the muscle cells and neurons (*Figure 6A*). Interestingly, as regeneration proceeded, *FoxA* expression in the epithelial cells diminished and became more pronounced throughout the interior part of the pharynx (*Figure 6B*), which is enriched for muscle and neurons. This expression data suggests that *FoxA* transcripts are present in several subtypes of pharyngeal tissue during regeneration and that expression may diminish once the organ is restored.

Neoblasts produce all cell types required to replace missing or dying tissues during normal tissue homeostasis and in response to amputation (*Pellettieri and Sánchez Alvarado, 2007*; *Reddien, 2011*; *Rink, 2012*). One model for how this is accomplished is that upon amputation, cells direct their output into the particular lineages that need to be regenerated. Based on the expression of *FoxA* in the stem cells, we asked whether it was required for the early steps in regeneration, including maintenance of the neoblast population and stimulation of local proliferation (*Figure 2*). Examination of the pattern of *Smedwi-1⁺* neoblasts showed that the overall distribution of stem cells was comparable in *FoxA(RNAi)* and control animals (*Figure 6C*), demonstrating that the neoblasts were not detectably affected by knockdown of *FoxA*. Consistent with this observation, body-wide and local proliferation during regeneration was indistinguishable from controls (*Figure 6D,E*) indicating that FoxA is not required for stimulation of proliferation.

The finding that *FoxA* is dispensable for proliferation highlights a key question about the phenotype. If neoblasts divide normally but fail to produce a pharynx, what is the ultimate fate of the division progeny produced by the stem cells? One possibility is that *FoxA* expression in the stem cells is required to direct the division progeny into pharyngeal fates. Interestingly, approximately 3 weeks after initiating RNAi, a pronounced dorsal outgrowth formed over the pharynx in 70% of animals (n > 100). Histological analysis of *FoxA(RNAi)* animals (*Figure 6—figure supplement 1*) demonstrated that this dorsal outgrowth lacked any recognizable anatomical features of control pharynges, such as radial symmetry or clear laminar structure.

Consistent with this observation, these dorsal outgrowths also failed to express either of two pharynx-specific markers (*Smed-laminin* and *Smed-npp-1*) (*Collins et al., 2010*) or the intestine-specific

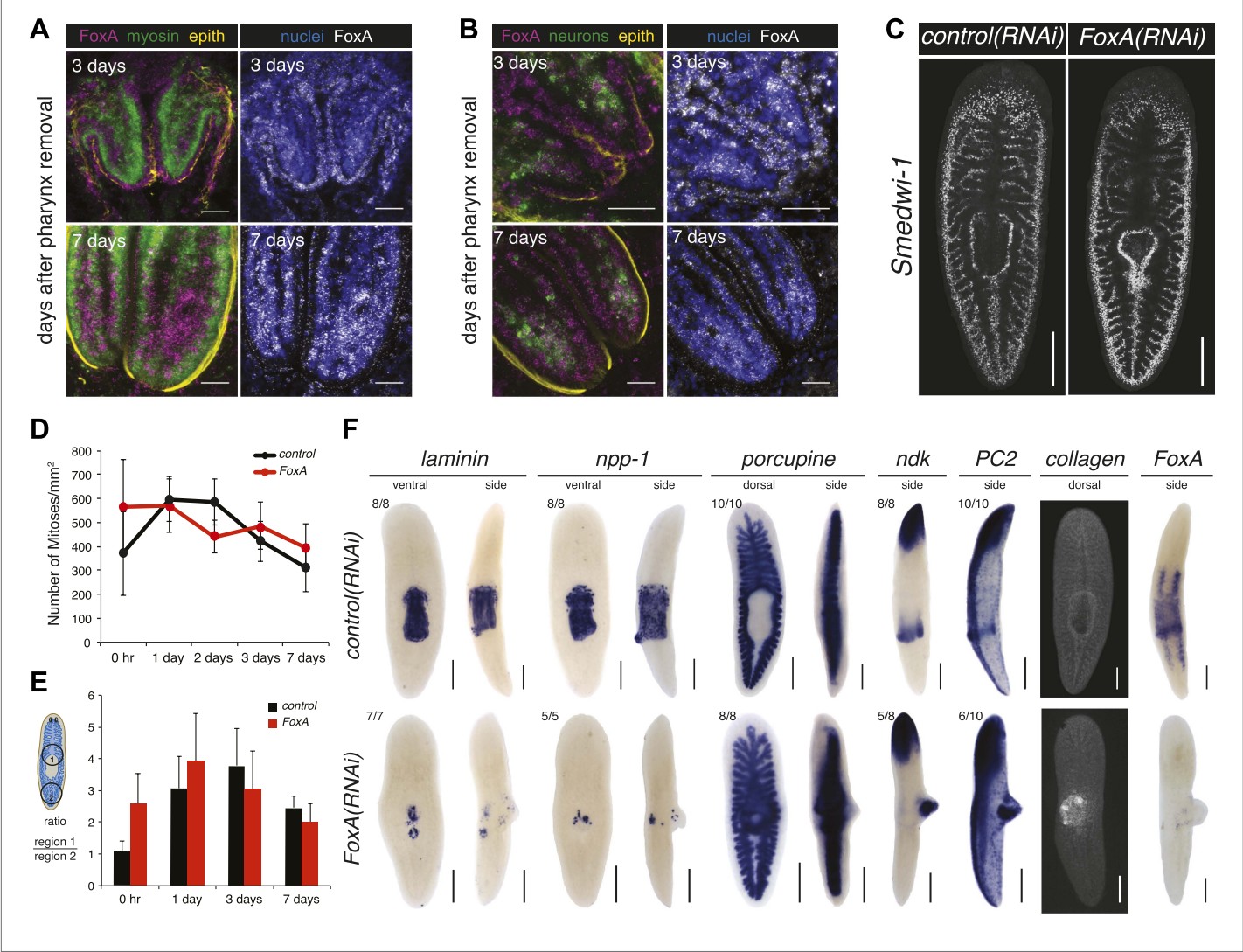

**Figure 6**. FoxA functions as a master regulator of the pharyngeal lineage. (**A** and **B**) Confocal images of animals stained for *FoxA*, *myosin* (muscle), *PC2* (neurons), α-acetylated tubulin (epithelial cells) and nuclei showing *FoxA* enrichment in epithelial cells 3 days after pharynx amputation, and shifting to mesenchyme 7 days after amputation. (**C**) Confocal images of *Smedwi-1* FISH (7 days after pharynx amputation) showing distribution of stem cells. (**D**) Body-wide phosphoH3-Ser10 staining in *FoxA(RNAi)* animals during pharynx regeneration. Error bars = SD. (**E**) Local phosphoH3-Ser10 staining during pharynx regeneration in *FoxA(RNAi)* animals. Error bars = SD. (**F**) Dorsal outgrowths in *FoxA(RNAi)* animals (day 20) lack pharyngeal tissue. Tissue-specific markers include: *laminin*, *npp-1* (pharynx), *porcupine* (intestine), *ndk*, *PC2* (neurons), *collagen* (muscle). In all images, anterior is up; in side views, dorsal is to the right. Scale bars (**A** and **B**) 50 μm, (**C**) 500 μm, (**F**) 250 μm.

The following figure supplements are available for figure 6:

**Figure supplement 1**. Dorsal outgrowths in *FoxA(RNAi)* are disorganized.

marker *Smed-porcupine* (*Figure 6F*), indicating that this aberrantly produced tissue does not acquire pharyngeal or endodermal fate. However, we did identify enrichment of neuronal markers (*Smed-PC2* and *Smed-ndk*) and the muscle marker *Smed-collagen*, which suggests that in the absence of FoxA, differentiation into neurons and muscle remains intact. To eliminate the possibility that this dorsal outgrowth is an incomplete pharynx formed by restoration of FoxA levels after the dsRNA inhibition has weakened, we examined *FoxA* expression 20 days after the final administration of RNAi, and found that it remained strongly suppressed (*Figure 6F*). Because these types of abnormal outgrowths were never observed in chemically amputated animals or in any other RNAi contexts, we conclude that FoxA likely acts to restrict differentiation of new tissues into the pharynx lineage.

## *FoxA* expression helps establish a molecular pathway for pharynx regeneration

FoxA appears to function specifically in a subset of stem cells to drive pharynx regeneration in adult animals. Based on its conserved functions controlling endoderm development in other organisms, we decided to use it as a landmark to dissect the function of novel genes uncovered in our screen and implicated in the pharynx regeneration pathway. Following amputation of the pharynx, *FoxA* is upregulated in the nascent pharynx and in the mesenchymal stem cells surrounding it (*Figure 5B*). We took advantage of this feature of FoxA to establish a molecular pathway for genes identified in our screen, reasoning that knockdown of genes functioning upstream of FoxA would inhibit accumulation of *FoxA*[+] progenitors, while genes functioning downstream of FoxA would not affect *FoxA* distribution. Indeed, failure to accumulate *FoxA*[+] cells was observed in irradiated animals (*Figure 5B*) and in *Smedwi-2(RNAi)* animals (*Figure 7A*), confirming that inhibiting stem cell function alters accumulation of *FoxA*[+] progenitors.

In situ hybridization of *FoxA* following RNAi knockdown of the remaining 19 candidate genes from our screen allowed clear categorization into distinct phases of pharynx regeneration. Based on the severe phenotypes of MCM7, ribonucleotide reductase, RuvB, cyclin D1, SAE-2, CPSF-3, and *zmym-1* in regeneration of head and tail structures after transverse amputation (*Table 1*), along with their enriched expression in stem cells (*Figure 3—figure supplement 1*), we had predicted that these

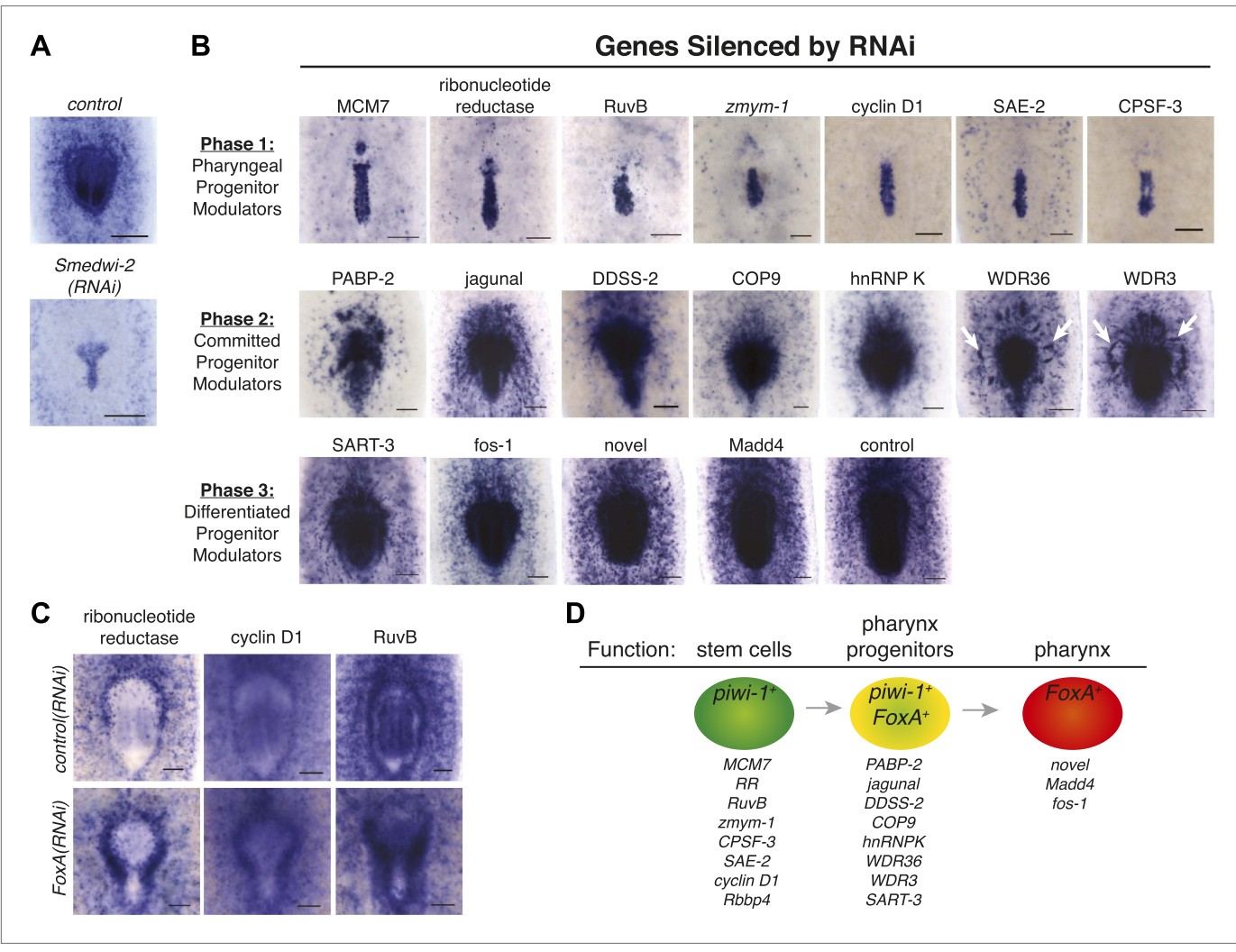

**Figure 7**. *FoxA* expression resolves a molecular pathway for pharynx regeneration. (**A**) *Smed-FoxA* expression 7 days after amputation in *Smedwi-2(RNAi)* animals. (**B**) *Smed-FoxA* expression 7 days after amputation following knockdown of the indicated genes. (**C**) Gene-specific in situ hybridization in *FoxA(RNAi)*. (**D**) Model for molecular control of pharynx regeneration. Scale bars = 100 µm.

genes were likely to act in stem cells. As expected, knockdown of these genes caused complete inhibition of *FoxA* progenitor accumulation, as well as a failure to maintain the mesenchymal population of *FoxA*+ cells (*Figure 7B*). Therefore, we conclude that this category of genes is required for the maintenance of pharyngeal progenitors, and confirmed that they function upstream of FoxA.

Knockdown of other genes caused significant, but less profound, defects in *FoxA* accumulation during pharynx regeneration. For example, *PABP-2(RNAi)* caused a severe decrease in the number of *FoxA*+ precursors accumulating in or around the pharynx (*Figure 7B*). Given the important role for translational regulation of planarian stem cell function (*Solana et al., 2013*), it is possible that PABP-2 (as well as CPSF-3) may act in stem cells or in their descendants. Knockdown of WDR3 and WDR36 caused an unusual distribution of *FoxA*+ precursors (*Figure 7B*, white arrows), with streams and clumps of *FoxA*+ cells present in the mesenchyme but directed toward the nascent pharynx. These phenotypes suggest the possibility that *FoxA*+ cells may be specified at a distance from the wound site, and then migrate towards it, and that WDR3 or WDR36 may function in either cell migration or maintenance of the pharynx regeneration program. Given the slight regeneration defects in head and tail regeneration caused by knockdown of WDR3 and WDR36 (*Table 1*), these genes may function broadly during regeneration of other organs. Knockdown of other genes (e.g., *Fos-1*, *Madd4*, and *SART-3*) caused subtle or undetectable defects in *FoxA* expression dynamics, suggesting that they function downstream of FoxA.

Our in situ hybridization timecourses demonstrated that expression of many candidate genes increased around the pharynx during regeneration (*Figure 3—figure supplement 1*). Because FoxA functions as a regulatory node in specifying pharynx regeneration, we analyzed the expression of candidate genes in *FoxA(RNAi)* animals to confirm their function either upstream or downstream of FoxA. This analysis was restricted to genes that showed clear expression changes in the vicinity of the pharynx during regeneration. Interestingly, ribonucleotide reductase, cyclin D1, and RuvB accumulated strongly around the nascent pharynx in *FoxA(RNAi)* animals (*Figure 7C*, white arrows), providing further evidence that in the absence of FoxA, stem cells are prevented from differentiating into the pharyngeal lineage.

## Discussion

Several major questions remain unanswered in planarian regeneration: how are stem cells stimulated to respond following amputation? How do stem cells differentiate into specific organs? How are newly regenerated organs integrated with the rest of the animal? To understand the molecular mechanisms driving these processes, we developed a method to selectively remove a single organ and combined this strategy with targeted molecular screens to identify key regulators of stem cell behavior and regeneration.

### Chemical amputation is an effective, reproducible and quantifiable paradigm for studying regeneration

Chemical amputation has several advantages over current surgical methods. First, the amount and types of tissues amputated is consistent among animals, stimulating the same regeneration program in each worm. Because planaria lack clear anatomical landmarks and vary in body proportion, surgical amputations are inherently variable among animals, removing different amounts of each tissue with each amputation. Second, chemical amputation produces wounds of exactly the same size, which normalizes both the extent of mitotic activity and the degree of apoptosis, both known to correspond directly with wound size (*Pellettieri et al., 2010*; *Wenemoser and Reddien, 2010*). Third, regeneration of the pharynx after chemical amputation can be quantified by measuring feeding behavior, facilitating rapid screens.

By reducing the complexity of amputation, we have simplified the challenge faced by neoblasts, requiring them to sense the absence of only one organ, and to channel their output into the pharyngeal lineage. This is in contrast to essentially all other types of surgical amputation performed in planaria, which introduce epithelial wounds and damage multiple underlying tissues that are broadly distributed throughout the body. Therefore, chemical amputation allows us to isolate the response of stem cells to the loss of a single organ in a potentially high-throughput, quantifiable manner.

### Expression profiling combined with functional analyses identifies genes acting in different stages of pharynx regeneration

Utilizing chemical amputation as a foundation for an RNAi screen, we were able to identify 20 genes that are required at multiple stages for pharynx regeneration, as measured by feeding behavior. However, given the small size of this screen, we have probably uncovered only a portion of the genes acting at each stage of pharynx regeneration. For example, two of these genes (Rhomboid and PDZ

ring finger 4) were undetectable immediately after amputation, but strongly upregulated 12–24 hr later (*Figure 2—figure supplement 3B*). Even though these genes failed to produce an RNAi pheno-type, their specific transcriptional activation after wounding indicates that mechanisms exist that allow animals to distinguish between homeostatic versus regenerative events. Nonetheless, we uncovered a pharyngeal regeneration molecular pathway for genes identified in this screen (*Figure 7D*).

Pharynx regeneration begins with activation of stem cells, an increase in expression of a pharynx-specific progenitor marker, and migration of these progenitors to the blastema. Some of the basic mechanisms driving regeneration (e.g., stem cell proliferation) are likely shared between the pharynx and other organs, and we identified several genes in this category, including ribonucleotide reductase, MCM7, and CPSF-3. In addition, we expected to identify genes specific to pharynx regeneration, and we identified at least one gene (*FoxA*) that appears to be highly specific for this organ. Interestingly, in *C. elegans*, the DNA helicase RuvB functions in a genetic pathway with FoxA to regulate pharynx organogenesis (*Updike and Mango, 2007*). Our results demonstrate that in planaria, RuvB is required for stem cell function, but it may also play a role in properly specifying progenitors during regeneration. A further category of genes identified were those that potentially affect the migration of progenitors into the nascent pharynx (WDR3 and WDR36), highlighting the fact that migration of stem cells and/or their progeny is an essential step in pharynx regeneration.

## Embryonic factors are deployed in adult animals to regulate pharynx regeneration

FoxA functions as a pioneer transcription factor, opening chromatin due to its structural similarity to linker histones and activating transcription of endoderm-specific genes (*Cirillo et al., 2002*; *Gaudet and Mango, 2002*; *Eeckhoute et al., 2009*). Although it has been previously characterized in development and cancer (*Lupien et al., 2008*), our study represents the first indication that FoxA functions in regeneration. FoxA is known to define organ identity during development (*Gaudet and Mango, 2002*) and to directly modify chromatin structure (*Cirillo et al., 2002*), leading to a hypothesis that during planarian regeneration, FoxA plays an analogous role. Because *FoxA(RNAi)* animals still established expression of anterior/posterior patterning molecules and regenerated head and tail structures normally, we conclude that FoxA is not required for regenerating or patterning organs besides the pharynx. We note, however, that some expression of pan-pharyngeal markers is maintained in *FoxA(RNAi)* animals, raising the possibility that organ specification is only partially compromised. Although this may be due to incomplete knockdown of *FoxA*, alternatively, it may reflect a block in differentiation of pharyngeal tissue, which requires FoxA activity.

The pharynx consists of neurons, muscle, mesenchyme, and epithelial cells, but lacks neoblasts. Each of these differentiated tissues has a distinct morphology and identity from the rest of the animal, suggesting that these different cell types may share a common pharyngeal identity. Based on our results, it is possible that FoxA may act to initiate the pharynx differentiation hierarchy to establish organ identity during regeneration, with additional layers of cell-specific differentiation occurring later. *FoxA* mRNA was expressed in multiple tissue types as regeneration progressed, suggesting that *FoxA* activation in stem cells is the first step toward differentiation of several pharynx-specific cell types. However, the exclusion of stem cells from the pharynx indicates that a boundary within the mesenchyme prevents pluripotent stem cells from invading this organ. *FoxA* expression bridges this spatial boundary.

## *FoxA* defines a subpopulation of stem cells necessary for pharynx regeneration

Our data demonstrate that *FoxA* transcript is present in the stem cell population, like other transcription factors that are critical for brain, photoreceptor, and excretory system development in planaria (*Lapan and Reddien, 2011*; *Scimone et al., 2011*; *Cowles et al., 2013*). However, in the case of pharynx regeneration, we can monitor the percentage of $FoxA^+$ stem cells in response to complete organ amputation, demonstrating that the stem cell population alters its output in response to organ amputation. The patterning and regeneration that occurs after amputation implies that signaling within the animal provides instructive cues guiding neoblast differentiation into particular fates. An interesting question raised by this work is how *FoxA* expression is triggered in stem cells. In other animals, FoxA recruitment to chromatin is controlled by trans-acting factors including T-Box, GATA, and lef transcription factors (*Mango, 2009*), and these types of proteins may function cell-autonomously in neoblasts.

Alternatively, signals acting distantly from the neoblasts are likely to stimulate *FoxA* expression in neoblasts. One possibility is that these signals originate in the pharynx, and normally limit the

production of *FoxA⁺* stem cells. These kinds of molecules, known as chalones and best typified by myostatin/GDF11 (***Bullough, 1965***; ***McPherron et al., 1997***), have been characterized in mammalian muscle and are thought to limit organ size in adult animals. In planarians, induction of supernumerary pharynges or engraftment of transplanted pharynges only occurs at a distance from the resident pharynx, suggesting that an inhibitory activity is present in the peripharyngeal region (***Ziller-Sengel, 1967a***, ***1967b***; ***Schilt, 1972***). Gaining a molecular handle on these aspects of whole-body regeneration will enhance our understanding of organismal homeostasis in animals.

In sum, our findings uncovered a new role for FoxA in adult animal regeneration and demonstrates that our organ-selective screening strategy can identify genes with distinct and specific functions during regeneration.

## Materials and methods

### Chemical amputation and tricaine anesthetization

Pharynges were removed from animals 4–5 mm in size, and starved for 7 days. Planarian water was replaced with 100 mM sodium azide (diluted in Montjuïc water). After 5–7 min, the pharynx was visibly extended out of the body. Vigorous pipetting often dislodged the pharynx from the body; if necessary, fine serrated forceps (#5441; Ted Pella, Reading, CA) were used to remove the pharynx, followed by several washes in Montjuïc. For tricaine treatment, 2 g/l tricaine was diluted in 10 mM Tris pH 7.5. Animals soaked in tricaine display their pharynx but never eject it.

### Worm care and irradiation

*Schmidtea mediterranea* asexual clonal line CIW4 was maintained and used as previously described (***Newmark and Sánchez Alvarado, 2000***). Animals were exposed to 6000 or 10,000 rads on a GammaCell 40 Exactor irradiator.

### Plug isolation, RNA extraction and microarray

At specified times after amputation, plugs were extracted using 1 mm microcapillary pipets (FHC, catalog # 30-30-0, Bowdoin, ME), and transferred directly into Trizol (Life Technologies, Grand Island, NY) using a mouth pipet. For each replicate, 25 plugs were homogenized together, and then chloroform-extracted. The pellet was then precipitated with isopropanol, washed, and resuspended in water. RNA was then purified on an RNEasy column with DNase-treatment (Qiagen, Germany).

The experiment was performed in triplicate. RNA quality was assessed on a Bioanalyzer 2100 machine (Agilent, Santa Clara, CA). Starting with 100 ng total RNA, amplification and labeling with Cy3 or Cy5 was performed using the Low Input Quick Amp Labeling Kit Two-Color from Agilent Technologies (#5190-2306). Custom Agilent 4x44k arrays with design id: 033226 were hybridized according to the manufacturer protocols, and scanned on an Agilent G2505C scanner. Data was analyzed in the R environment using the Limma library (***Smyth, 2004***) for loess normalization and calculation of p-values between treatments. p-values were adjusted for multiple hypothesis testing by the method of (***Benjamini and Hochberg, 1995***). The data have been deposited in GEO with accession number: GSE56181.

### Cloning, RNAi screening, and feeding assay

Primers with overhangs homologous to pPR-T4P vector (J Rink) were used for PCR amplification from a cDNA library generated with SuperScriptIII (Life Technologies). PCR products were treated with T4 polymerase, mixed with linearized vector (digested with *SmaI* and treated with T4 polymerase) and incubated for 15 min at room temperature. Ligations were transformed directly into *Escherichia coli* strain HT115, then verified by PCR and sequencing. For screening, overnight cultures of individual cDNAs were grown in 2XYT, and 2X RNAi food was prepared (30 ml bacterial culture was pelleted and resuspended in 150 µl 3:1 liver:water paste). 15 animals were fed three times, 3 days apart, and amputations were done the following day. Feeding assays were performed 9 days after amputation. All RNAi experiments used this same timing strategy, with day 0 representing the time of amputation. Sequences of genes used in this study are deposited in GenBank with accession numbers KJ573350-KJ573369.

For the feeding assay, animals were transferred into a new petri dish and kept in the dark for at least an hour. Diluted liver paste consisting of approximately 4:1 liver:planarian water and 20 µl red food coloring was mixed and 25 µl was pipetted into the petri dish. Percentage of animals with red intestines were scored after approximately 30 min food exposure. For *FoxA(RNAi)*, we used 4X RNAi food (100 ml overnight culture resuspended in 250 µl liver paste).

## In situ hybridizations, immunohistochemistry, and image quantification

In situ hybridizations used the protocol in *Pearson et al. (2009)* for colorimetric development and the protocol in *King and Newmark (2013)* for fluorescent development except that animals were fixed for 45 min in a solution containing 4% PFA, 0.5% Triton X-100, and 1X PBS. For mounting, we soaked fluorescently stained animals overnight in modified ScaleA2 solution for improved optical clarity (*Hama et al., 2011*) containing 40% glycerol, 2.5% DABCO (Sigma–Aldrich, St. Louis, MO) and 4M urea. For cryosectioning and immunohistochemistry, after completion of WISH animals were fixed overnight at 4°C in 4% paraformaldehyde (in PBS), washed three times in PBS, equilibrated in 30% sucrose, frozen in OCT and cryosectioned (10 µm thick). To stain sections, slides were incubated in 1X Powerblock (Biogenex, Fremont, CA) for 30 min, then incubated with rabbit monoclonal acetylated tubulin at 1:1000 (#5335; Cell Signaling Technology, Danvers, MA) and Tmus (kind gift of Rafael Romero, used 1:1000) for at least 1 hr, and developed with Alexa-conjugated secondary antibodies diluted 1:1000 (Abcam, Cambridge, MA). All antibodies were diluted in Antibody Diluent Solution (Life Technologies). Anti-H3Ser10Phos (Millipore, Billerica, MA) was used at 1:1000 and developed with Alexa-conjugated goat anti-rabbit secondary antibodies (150086; Abcam).

For hematoxylin and eosin staining, animals were fixed overnight at 4°C in 4% paraformaldehyde (in PBS), then washed 3X in PBS and dehydrated by an ethanol series through washing in 30%, 50%, 70%, 80%, 95%, and 100% ethanol (7 min each). To embed in paraffin, animals were soaked in ethanol with 5% glycerol, washed in xylene (7 min) and clear-rite (2 × 7 min), and soaked in paraffin (2 × 14 min, then 2 × 30 min). After serial sectioning (10 µm thickness), slides were heated to 60°C for 20 min, deparaffinized with three 2-min washes in xylene, washed 3 × 1 minute in 100% ethanol, then 80% ethanol, rinsed in tap water, and then incubated 30 s in hemalast, 2 min in hematoxylin, rinsed 2 min in tap water. Staining utilized the Leica Infinity system and was performed in a Leica Autostainer.

Colorimetric WISH images were captured on either Zeiss Lumar or Leica M205 stereoscopes. Confocal images were captured on a Zeiss LSM510-VIS inverted microscope with a 20X or 40X objective. For quantification of phosphohistone H3, full slide or individual worm tiled image sets were acquired on a Perkin Elmer Ultraview spinning disk microscope. Stitching was performed using stitching plugins in FiJi with customized batch processing macros or wrapper plugins where necessary. Custom plugins were used to segment the DAPI labeled worms and the 'Find Maxima' function was used to count spots, both wrapped in batch processing macros. All macros and plugins are available at https://github.com/jouyun.

## Acknowledgements

We thank Hanh Vu, Jochen Rink, and Li-Chun Cheng for technical advice and critical suggestions throughout this project, and all other members of the Sánchez Lab, past and present, for advice along the way. We would also like to acknowledge technical support from members of the Histology, Microscopy, and Molecular Biology Cores at the Stowers Institute. This work was supported by NIH Developmental Biology Training Grant 5T32 HD07491 and NRSA F32GM084661 (CEA), and NIH R37GM057260 to ASA. ASA is an investigator of the Howard Hughes Medical Institute and an Investigator of the Stowers Institute for Medical Research.

## Additional information

### Competing interests

ASA: Reviewing editor, *eLife*. The other authors declare that no competing interests exist.

### Funding

| Funder | Grant reference number | Author |
| --- | --- | --- |
| National Institutes of Health | NIH R37GM057260 | Alejandro Sánchez Alvarado |
| Howard Hughes Medical Institute | N/A | Alejandro Sánchez Alvarado |
| Stowers Institute for Medical Research | N/A | Alejandro Sánchez Alvarado |

The funder had no role in study design, data collection and interpretation, or the decision to submit the work for publication.

## Author contributions
CEA, Conception and design, Acquisition of data, Analysis and interpretation of data, Drafting or revising the article; CWS, Temporal expression data analyses, Acquisition of data, Analysis and interpretation of data; SAM, Data imaging and quantification, Acquisition of data, Analysis and interpretation of data; ASA, Conception and design, Analysis and interpretation of data, Drafting or revising the article

## Additional files

### Supplementary files
• Supplementary file 1. (**A**) Complete list of genes screened. (**B**) Genes used for whole-mount ISH validation of microarray.

### Major dataset

The following dataset was generated:

| Author(s) | Year | Dataset title | Dataset ID and/or URL | Database, license, and accessibility information |
|---|---|---|---|---|
| Adler CE, Seidel CW, McKinney S, Sánchez Alvarado A | 2014 | Selective amputation of the pharynx identifies a FoxA-dependent regeneration program in planaria | http://www.ncbi.nlm.nih.gov/geo/query/acc.cgi?acc=GSE56181 | Publicly available at NCBI Gene Expression Omnibus. |

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
