## [Decision Letter]

Thank you for sending your work entitled “Selective amputation of the pharynx identifies a FoxA-dependent regeneration program in planaria” for consideration at *eLife*. Your article has been favorably evaluated by a Senior editor, Janet Rossant, a Reviewing editor, Marianne Bronner, who is a member of our Board of Reviewing Editors, and 2 reviewers.

The Reviewing editor and the other reviewers discussed their comments before we reached this decision, and the Reviewing editor has assembled the following comments to help you prepare a revised submission.

This manuscript by Adler et al. describes the results of an elegant screen for genes required for organ regeneration in planaria. The authors have developed a clever scheme to chemically amputate the planarian pharynx, thereby standardizing the organ's removal and permitting regeneration assays to be conducted on a relatively large scale. These assays have a straightforward, quantitative readout: animals that regenerate a functional pharynx are able to ingest food (dyed to monitor food intake). The authors characterize the cell biology underlying pharynx regeneration, showing that it requires the animal's stem cells, and they identify changes in gene expression that occur during the initial three days of pharynx regeneration. Targeting a subset of genes that are differentially expressed during the early stages of pharynx regeneration, the authors perform an RNA interference screen to identify genes required for this process. Of the 356 genes they screened, they find 20 in which knockdown leads to defects in pharynx regeneration, by perturbing either stem cell function, pharyngeal morphogenesis, or more specific aspects of pharyngeal function (i.e., a pharynx is formed, but it does not function properly). Focusing their efforts on a forkhead transcription factor homolog, FoxA, the authors show that this gene is required specifically for regeneration of the pharynx. In addition to its expression in the pharynx, FoxA is expressed in a subset of *smedwi-1+* neoblasts, providing additional evidence for lineage commitment within this cell population.

This elegant work provides an important contribution to our understanding of organ regeneration in planaria. The paper is well written, the data are beautifully presented, and the work is consistent with the high standards that one has come to expect from an *eLife* paper. I recommend that it be accepted, provided that the following relatively minor concerns are addressed.

Substantive concerns that need to be addressed:

1) The interpretation that the dorsal outgrowths observed 3-4 weeks after initiating Fox RNAi treatment are not specified as pharyngeal tissue seems problematic given the analysis reported in Figure 6. Because the only markers shown are general markers of protonephridia and neurons (i.e., no pharyngeal markers are shown), the identity of these outgrowths appears to be unresolved. If FoxA expression begins to recover at these late time points (i.e., the effects of dsRNA wear off), these outgrowths could reflect defective/aborted pharyngeal regeneration based upon insufficient FoxA levels. This could easily be addressed by examining the outgrowths for the expression of pharyngeal markers (and FoxA, to see if it is still inhibited).

2) Although it is nice to refer to Morgan's and Randolph's papers from the late 1800s, it would seem that there has been enough recent progress on planarians and sufficient historical reviews referring extensively to older work to render these citations superfluous. Instead, it seems more important to cite relevant work from this and other post-Morgan/Randolph eras. For example, it would be reasonable to cite the ultrastructural work on the pharynx from the 1960s. The discussion of chalones at the end of the paper would benefit from considering the work of Ziller-Sengel and Schilt ('60s and '70s), suggesting the presence of an autoinhibitory factor from pharynx. The paper characterizing TMUS-13 (Cebria et al. 1997) should be referenced, since that antibody is used extensively.

---

## [Author Response]

*1) The interpretation that the dorsal outgrowths observed 3-4 weeks after initiating Fox RNAi treatment are not specified as pharyngeal tissue seems problematic given the analysis reported in*
Figure 6*. Because the only markers shown are general markers of protonephridia and neurons (i.e., no pharyngeal markers are shown), the identity of these outgrowths appears to be unresolved. If FoxA expression begins to recover at these late time points (i.e., the effects of dsRNA wear off), these outgrowths could reflect defective/aborted pharyngeal regeneration based upon insufficient FoxA levels. This could easily be addressed by examining the outgrowths for the expression of pharyngeal markers (and FoxA, to see if it is still inhibited)*.

The reviewers make a valid point and we agree that the analyses presented in the first submission were insufficient. In the revised version, we have added several WISH experiments interrogating the cellular composition of this dorsal outgrowth. In particular, our experiments confirmed the absence of pharyngeal tissue using two specific markers (*laminin and npp-1*) in all animals examined. By H&E staining, the dorsal outgrowth lacks any apparent radial or laminar structure characteristic of the intact pharynx. Often, this outgrowth is densely populated with muscle and neuronal tissue. Furthermore, FoxA expression remains strongly inhibited at these late time points, eliminating the possibility that restoration of FoxA expression drives formation of this outgrowth. We have included this data in Figure 6 and in Figure 6—figure supplement 1.

*2) Although it is nice to refer to Morgan's and Randolph's papers from the late 1800s, it would seem that there has been enough recent progress on planarians and sufficient historical reviews referring extensively to older work to render these citations superfluous. Instead, it seems more important to cite relevant work from this and other post-Morgan/Randolph eras. For example, it would be reasonable to cite the ultrastructural work on the pharynx from the 1960s. The discussion of chalones at the end of the paper would benefit from considering the work of Ziller-Sengel and Schilt ('60s and '70s), suggesting the presence of an autoinhibitory factor from pharynx. The paper characterizing TMUS-13 (Cebria et al. 1997) should be referenced, since that antibody is used extensively*.

Agreed. We have added the suggested references.